# En-Bloc Resection of Metastases of the Proximal Femur and Reconstruction by Modular Arthroplasty is Not Only Justified in Patients with a Curative Treatment Option—An Observational Study of a Consecutive Series of 45 Patients

**DOI:** 10.3390/jcm9030758

**Published:** 2020-03-11

**Authors:** Oliver E. Bischel, Arnold J. Suda, Paul M. Böhm, Burkhard Lehner, Rudi G. Bitsch, Jörn B. Seeger

**Affiliations:** 1BG Trauma Center at University of Heidelberg, Ludwig-Guttmann-Str. 13, 67 071 Ludwigshafen, Germany; 2AUVA Trauma Center Salzburg, Department of Orthopedics and Trauma Surgery, Academic Teaching Hospital of the Paracelsus Medical University, Dr.-Franz-Rehrl-Platz 5, 5010 Salzburg, Austria; arnoldsuda@yahoo.com; 3General Orthopedics, Neumeyerstr. 46, 90 411 Nuremberg, Germany; info@ortho-boehm.de; 4Department of Orthopedics and Traumatology, University of Heidelberg, Schlierbacher Landstr. 200a, 69 118 Heidelberg, Germany; Burkhard.Lehner@med.uni-heidelberg.de; 5Atos Clinic, Bismarckstraße 9-15, 69115 Heidelberg, Germany; Rudi_Georg.Bitsch@urz.uni-heidelberg.de; 6Parc Clinic, Am Kaiserberg 2-4, 61 231 Bad Nauheim, Germany; joernseeger@gmx.net

**Keywords:** proximal femur, secondary bone tumor, megaprosthesis, survival of implant, survival of patients

## Abstract

Background: There is little conformity regarding the surgical treatment of metastasis of the proximal femur, especially in palliative patients with limited life expectancy. Patients and Methods: En-bloc resection of secondary bone malignancies of the proximal femur and reconstruction by modular arthroplasty was performed in a consecutive series of 45 patients. The mean follow-up period was 16.4 months (0.6–74.7). Results: The survival rate of all patients was 6.6% (95% CI: 0–14.9) at 74.7 months. There was no significant difference in patients with a solitary or disseminated disease at index operation (log-rank *p* = 0.1214). Recurrent dislocation was the most frequent local complication (*n* = 6) necessitating an open reduction in four cases. The use of a Trevira tube showed a higher risk of dislocation compared to the simple bonding of remaining soft tissue (6 out of 28 vs. 0 out of 17; Fisher test: *p* = 0.0463). The worst-case survival rate with the removal of the arthroplasty for any cause and/or loss to follow-up was 80.0% (95% CI: 44.9–100) at 74.7 months (*n* = 1 due to low-grade infection). Conclusions: En-bloc resection of metastases and reconstruction by modular arthroplasty is reliable even in patients with very limited life expectancy. Local complications due to tumor growth or instability after intralesional surgery could be managed successfully but recurrent dislocation as the most frequent complication has to be taken into account. The simple bonding of remaining soft tissue around the prosthesis without the use of an attachment tube may reduce the dislocation rate and reoperation risk.

## 1. Introduction

Frequent malignancies like myeloma, kidney, breast or prostate cancer are known to have a high incidence of developing bony metastasis during the disease and the survival of these patients has improved steadily due to targeted treatment options. The proximal femur is the most frequent localization for bone metastases besides the spine. The risk of pathological fractures at the proximal femur is high due to weight-bearing and biomechanical conditions [1,2,3]. Osteosyntheses by a locking plate, dynamic hip screw (DHS) or intramedullary nail, with or without cement augmentation, are still used most frequently for palliative treatment. Despite additional radiotherapy complications following intralesional procedures like persistent pain due to instability and local tumor progression, especially at the proximal femur, may often appear [4,5,6]. Overall, there seem to be higher implant failure rates after osteosynthesis compared to endoprosthetic reconstruction already within the first year, further increasing with longer survival [7,8].

In contrast to intralesional procedures, en-bloc resection of the metastasis and endoprosthetic reconstruction by modular devices offers the possibility to prevent local complications and can be performed for both curative and palliative treatment [9,10,11].

Nevertheless, surgical treatment of metastasis, especially in palliative patients, is discussed controversially. It includes less invasive treatment by intralesional procedures as well as aggressive en-bloc resection with wide margins. Consecutively, the variety of implants ranges from osteosynthesis to modular tumor systems.

Less data is available for a larger series of patients treated operatively with en-bloc resection of bone metastasis of the proximal femur and reconstruction with a modular tumor system. The aim of this study was to evaluate whether aggressive en-bloc resection and reconstruction by modular tumor prosthesis are effective and reliable even in advanced tumor stages. Survivorship data of the implant, clinical outcome and complication rates were therefore analyzed in a retrospective study with respect to a patient’s survival in this cohort.

## 2. Patients and Methods

A consecutive series of 45 patients with en-bloc resection of metastasis of the proximal femur and reconstruction with a modular tumor endoprosthesis between 1997 and 2010 was evaluated retrospectively.

### 2.1. Inclusion Criteria and Surgical Algorithm

All patients were treated according to a vote after presentation at an interdisciplinary tumor board preoperatively. Patients with the decision of surgery were treated with respect to individual patient’s specifics according to the following surgical algorithm. Inclusion criteria for the current cohort with en-bloc resection and reconstruction with a modular tumor device were the presence of a tumor at the proximal femur with extension distally to the intertrochanteric line. Additional postoperative radiotherapy was planned in patients with intralesional procedures dependent on the pathological status of the specimen. As survival of the patients was not significantly different for patients with a solitary and disseminated disease we included all cases.

In patients presenting a fracture or at a high risk of fracture (Mirels score of at least 9) and very poor prognosis of less than approximately 3–6 months, either a simple osteosynthesis with use of an intramedullary nail was performed when the tumor was located below the intertrochanteric zone (meta- and/or diaphysis) and the region of the femoral neck was intact. In localizations proximal to the intertrochanteric line and/or femoral neck region and in tumors with a good response to radiotherapy, a cemented primary stem after curettage was used.

### 2.2. Indication for Surgery and Methods

26 patients were female and 19 male with a total of 22 surgical procedures on the left side and 23 on the right. At the time of surgery, the mean age was 58.7 (34.0–85.0) years. The mean follow-up period was 16.4 (0.6–74.7) months and data was available for all cases (*n* = 45).

The entity of the underlying malignancy, staging at the time of index operation and indication for operation are listed in Table 1 and Table 2. A metastasis of the proximal femur had been stabilized operatively before the index operation in 5 patients. A failed osteosynthesis was causative for index surgery in four and a total hip arthroplasty (THA) in one patient (Figure 1 and Figure 2). The mean Mirels score of the patients presenting with impending fractures (*n* = 23) was 9 (7–11) points.

All patients were evaluated clinically and radiologically. Pre and postoperative activity and general condition were measured using the Karnofsky index [12]. Function was evaluated according to the MSTS score [13]. Local complications like infection or dislocation with or without revision or consecutive implant failure were collected and statistically evaluated. Migration analysis was performed according to the technique proposed by Callaghan et al. [14].

Statistical analysis was performed with JMP 10 for MAC (SAS Institute Inc., Cary, NC). A time-to-event analysis was performed using the Kaplan–Meier method with death, removal of the stem for any cause, aseptic loosening of the stem and worst case (removal of the stem for any cause and/or aseptic loosening and/or lost to follow up) and dislocation as failure criteria. A 95% confidence interval was applied to all survivorship data and *p*-values for comparing survival curves were calculated with the log-rank test. One-, three- and five-year survival data were calculated for patient survival. Associations or correlations between continuous and/or discrete variables were tested by the Fisher, Student’s t-, Paired t- or Chi square-test depending on the underlying empirical distribution. All tests were two-sided and *p* ≤ 0.05 was considered significant.

### 2.3. Surgical Technique and Postoperative Care

A modular arthroplasty system for reconstruction of the proximal femur was used (MUTARS^®^, Implantcast, Buxtehude, Germany). A Trevira tube for protection of the dislocation and refixation of the muscles at the implant was applied in 28 patients. In 17 remaining cases without a tube, the tendons of the pelvitrochanteric muscles were intact and could be attached to the quadriceps (vastus lateralis, intermedius and/or rectus) or in the case of a resected vastus lateralis, to the maximus insertion (dorsal parts) and/or fascia. The psoas tendon was fixed around the prosthesis by non-resorbable sutures. To protect the dislocation of bipolar heads the capsule of the hip joint was preserved, and retention sutures were sewn for a tight closure of the capsule after repositioning the prosthesis.

Bone resection length was assessed during surgery by X-ray validation according to the preoperative planning. The mean resection length of the proximal femur was 14.5 (7–25) cm with a median of 14 cm. The histopathological outcome is described in Table 3. The mean reconstruction length of the prosthesis measured 14.4 (7.6–25.6) cm with a median of 14.0 cm. A bipolar head was inserted in 32 operations. In 12 cases an acetabular component was implanted; 11 of them were treated by a cemented PE cup, one with a cementless screw-in cup. The stable shell of the patient with a pre-existing THA was left in situ (Figure 1 and Figure 2).

Pre- and/or postoperative radiotherapy of the tumor region occurred in 22 patients and pre- and/or postoperative radio- and/or chemotherapy in 28 patients. Four out of eight patients received postoperative radiotherapy after intralesional surgery.

## 3. Results

### 3.1. Complications and Reoperations

Two patients died within four weeks after surgery due to tumor-related general weakness. One hip joint was removed due to low-grade infection 34.5 months, postoperatively. In one patient, the revision of postoperative hematoma with additional debulking of the left soft-tissue tumor mass was performed. Open reduction was necessary in four patients, one with a bipolar head and three with a THA. Consecutively, the cumulative risk of revision without removal of the prosthesis (*n* = 5) was 11.6% at 74.7 months.

A total of 2 out of 32 hips with bipolar heads and four of the 13 THAs dislocated. Recurrent dislocation was more likely in patients with severe soft tissue resections treated additionally by a Trevira tube than in patients with simple suture of remaining muscles and tendons as described earlier (6 out of 28 vs. 0 out of 17; Fisher test: *p* = 0.0463). Dislocation was more likely in total hip arthroplasties (Fisher test: *p* = 0.0488) with a significantly higher cumulative risk for dislocation compared to hemiarthroplasties (4 out of 13 with 31.6% at 49.5 months; 95% CI 5.8–57.4 vs. 2 out of 32 9.6% at 74.7 months; 95% CI 0.7–23.0; log-rank test: *p* = 0.0167; Figure 3). Resection/reconstruction length (Figure 4 ) or local radiotherapy showed no influence on the risk of dislocation.

Cumulative risk of reoperation without removal of the prosthesis (*n* = 6) was 11.6% at 74.7 months. A temporary lesion of the peroneal nerve with complete remission was observed in one patient.

### 3.2. Functional Evaluation

Before full weight-bearing was reached, three patients died. The direct postoperative data of these patients were included in the functional scores. The mean Karnofsky index improved from 49.4 (20%–90%) preoperatively to 54.9 (10%–90%) and the MSTS score from 26.4 (0.0%–96.7%) preoperatively to 51.3 (6.7%–93.3%). The mean pain score improved from 1.1 (0–5) points preoperatively to 3.9 (0–5) points postoperatively. Of the 45 patients, 28 had no or mild pain (four or five points) with NSAIDs as the only pain medication (Figure 5). The use of a Trevira tube did not influence the functional outcome. The MSTS score with its subitems especially the use of walking aids, walking ability and gait were not influenced significantly.

### 3.3. Radiographic Evaluation

No subsidence or migration of the stems was seen, and all stems remained stable until the last follow-up. Postoperatively, leg length discrepancy of more than 1 cm could be measured in nine patients with a mean difference of 0.2 (−2.5–2) cm.

### 3.4. Survival Analysis

Within six months after surgery, 16 patients died. Survival of all patients was 6.6% (95% CI: 0–14.9) at 74.7 months, six patients were alive at the last evaluation including one patient with implant failure. The one-year and three-year survival rates were 52.9% (95% CI: 38.2–67.6) and 13.2% (95% CI: 2.6–23.7). Patients with disseminated disease (*n* = 30) had an overall survival rate at 74.7 months of 3.5% (95% CI: 0–10.3). One-year and three-year survivorship rates were 45.7% (95% CI: 27.7–63.8) and 7.0% (95% CI: 0–16.4). The survival rate of patients with a solitary metastatic disease at the time of operation (*n* = 15) was 13.5% (95% CI: 0–36.0) at 57.0 months. One-year and three-year rates were 66.7% (95% CI: 42.8–90.5) and 27.0% (95% CI: 4.6–49.4), respectively. Nevertheless, a log-rank test showed no significance between survival rates of patients with solitary or disseminated disease (*p* = 0.1214) (Figure 6). Patients with impending (*n* = 24) or an apparent fracture (*n* = 21) showed a survival rate of 5.2% after 57.0 months or 8.4% after 74.7 months. Survival was not significantly different between the two groups (log-rank *p* = 0.8265).

The worst-case survival rate of hip arthroplasty was 80.0% (95% CI: 44.9–100) at 74.7 months with one removal of the stem 34.5 months postoperatively due to low-grade infection (Figure 7). No patient was lost to follow up.

## 4. Discussion

### 4.1. Background, Principle Considerations for En-Bloc Resection and Functional Outcome

Quality of life can be improved by the surgical intervention of bony metastases in most patients. Even in an advanced stage of tumor disease with a limited life expectancy of only a few months, patients may benefit from at least partial restoration of function [15]. En-bloc resection of solitary metastasis is well established and obligatory to pursue a curative regimen. However, aggressive surgery of a metastasis of the proximal femur in patients with disseminated disease is discussed controversially.

For decades, surgical treatment has not changed relevantly and less invasive procedures like intralesional stabilization by osteosynthesis or cemented endoprosthetic reconstruction after curettage are still performed most frequently. Only three decades ago, overall survival rates of patients presenting bony metastasis were very limited and, therefore, longer-lasting reconstructions were most often not necessary [16,17]. In contrast to surgical therapy, a patient’s survival of most entities has improved steadily due to a targeted systemic treatment during the same period.

Failure of osteosynthesis and/or arthroplasty after intralesional surgery of lesions of the proximal femur was an indication for surgery in this cohort in nine respectively 2% (osteosynthesis *n* = 4; arthroplasty *n* = 1). Failure occurred at a mean of four (0.1–8.6) months after osteosynthesis. Comparable studies have described a rate of collapse of osteosyntheses in the proximal femur of up to 23% within the first year [4,5,6,18]. Additionally, the risk for failure may increase with a patient’s survival.

Overall, there are significantly higher survival rates following arthroplasty compared to osteosynthetic stabilization at the proximal femur [5,10,19,20]. In contrast to osteosynthesis, reconstruction with arthroplasty offers the possibility of an extralesional (marginal or wide) resection of the tumor that is not only required in patients with a curative regimen in primary tumors or solitary metastatic disease but also in disseminated stages for local tumor control [9,10]. Additional postoperative radiotherapy is not necessary for tumor control or pain reduction. In patients with a limited life expectancy of less than three months, radiotherapy for pain reduction was ineffective in more than half of the patients [21]. Accordingly, marginal or wide resection may also be favorable for pain control especially for patients with longer life expectancy. This finding is similar to other studies and should especially be considered for tumors with poor response to radiotherapy like kidney cancer [4,22,23].

In this series, 9% had no or mild pain preoperatively (MSTS pain score of 4–5 points) but more than 62% postoperatively, which could be treated sufficiently by NSAID as a single medication. Nevertheless, the postoperative MSTS score of 51.3% was less compared to other series with scores of 67% to 87% [24,25,26,27]. Inclusion of patients with primary bone tumors in these studies and incorporation of many patients in an advanced stage in this series are limiting factors [1,24,25,26,27]. All data were included despite the death of 16 patients within six months after surgery. Although functional results were not significantly different between the two groups (follow-up greater or less than six months), differences in pain scores were less than in all other aspects of the MSTS score (pre- vs. postoperatively). At the last evaluation 10 out of the 16 patients (59%) with a limited life expectancy of less than six months had no or mild pain (more than four points) vs. 18 out of 28 of the group with longer follow-up (64%). Therefore, surgical resection of the metastasis and endoprosthetic reconstruction seems to be efficient for local pain control even in patients with a very limited survival.

Our study has limitations. The comparatively small number of patients (*n* = 45) is one disadvantage. Additionally, the follow-up period is limited due to the disease of the patients. Furthermore, a control group with patients treated by simple osteosynthesis, intralesional curettage and cemented primary stems or revision devices is missing. Nevertheless, there is adequate data available to compare the results of the current cohort and to draw the described conclusions. There are also limitations due to the study design as it is a retrospective study. Prospective randomized studies may be necessary in the future.

### 4.2. Complications and Survivorship of Patients and Implants

At current, survival rates of 90% at one year or 20% at five years are described in patients with favorable conditions with respect to prognostic factors of entities showing a high risk of developing bony metastases during follow-up [4,28,29]. After surgical treatment of bone metastases of the lung, prostate, and kidney or breast cancer, patient survival rates of 40% after one year and 20% after three years are reported [1]. In our series, the one-year, three-year and five-year survival rates were 52.9%, 9.9% and 6.6% respectively. Consecutively, this cohort consisted mainly of patients with a limited life expectancy but a survival of more than one year has to be considered in at least half of the patients.

Endoprosthetic treatment of patients with metastatic disease of the proximal femur in this series showed a survival rate of the reconstruction of 80.0 (44.9%–100%) at 74.7 months with one removal of the stem 34.5 months postoperatively due to low-grade infection. Of the few studies discussing arthroplasty in metastases of the proximal femur, comparable survival rates of the prosthesis at five years between 83% and 100% were reported [19,30].

Revision rates with the removal of the endoprosthesis after tumor reconstruction due to infection between 1.8% and 21% have been published [19,24,25,27,31,32,33] further increasing the patients treated by neo-/adjuvant, intra-operative or direct adjuvant radiotherapy [31]. There was one implant failure due to infection despite pre- and/or postoperative radio- and/or chemotherapy in 28 patients in our investigation.

Dislocation rates of megaprostheses of up to 34% have been described [9,27]. A dislocation rate of 13.3% (six hips) in this cohort is comparable with available data in tumor and revision surgery [5,6,26,34,35]. Nevertheless, instability is a major local complication in this series with a time-dependent risk of 16.2% at 74.7 months. As discussed in other studies a significantly higher risk could be seen in THA than hemiarthroplasties (Fisher test: *p* = 0.0488) [36]. None of the patients with attachment of intact soft tissue by single sutures dislocated while 6 out of 28 patients with deficient soft tissue and additional use of a tube presented instability during follow up. Nevertheless, clear data showing the improvement of dislocation rates by using certain devices for soft tissue refixation are missing [34,36,37,38]. The resection length of the proximal femur and consecutive implant-related parameters like reconstruction length did not influence the dislocation rate in this study. However, the extent of the extra-osseous tumor mass might influence and explain the dislocation rate. In extensive tumors with a large extra-osseous tumor mass involving the joint capsule and stabilizing muscles, successful soft tissue attachment is difficult to address.

There was no difference in the functional outcome (MSTS score with sub-items), which should be improved, as it is the other aim of using the tube due to tight fixation of the muscles at the prosthesis. In the case of dislocation, the tube may hinder closed reposition as it was the reason for two of four open reductions. Therefore, in cases with a preserved capsule as the best prevention for instability, we would not routinely recommend the use of a tube and, in our experience, its benefit is questionable. Consecutively, a time-dependent revision rate without removal of the prosthesis of 12.6% could be further reduced but seems to be acceptable and is lower compared to revision rates due to mechanical failure of osteosyntheses.

The use of primary stems or cemented long stems would have been an option in some cases [39]. From a biomechanical point of view, en-bloc resection of a highly insufficient proximal femur is identical to an intralesional procedure with a cemented primary or long-stemmed prosthesis. The length of anchorage in intact bone is more or less similar with both procedures. Due to these considerations and the topics mentioned above, the more aggressive treatment in this selective cohort was performed. A cementless Wagner SL revision stem was implanted in a similar group of patients with comparable results [22]. Obviously, both systems allow accurate therapy in these patients.

Our procedure in the operative treatment of secondary bone malignancies at the proximal femur can be summarized as follows:Simple osteosynthesis by intramedullary nail is justified in cases with a very poor prognosis of a few months (≈ less than six) and a tumor localization clearly below the intertrochanteric region (meta- and/or diaphysis).Intralesional curettage and implantation of a primary stem is justified in patients with a very limited life expectancy and a localization proximal to the intertrochanteric line/femoral neck in tumors with a good response to radiotherapy.Intralesional curettage and implantation of a long-stemmed, cemented prosthesis can be performed in patients with very limited life expectancy in biomechanical situations not suitable for osteosynthesis and/or a cemented primary stem and in patients with additional metastasis more distally.All other cases should be treated by en-bloc resection (wide or marginal) and endoprosthetic reconstruction.A cemented reconstruction should be performed in patients with necessary potential postoperative radiotherapy (R1, intralesional) and/or additional metastasis distally; all other cases can be treated with cementless devices.A tube for soft tissue attachment and prevention of dislocation is not necessary when the capsule can be preserved.


## 5. Conclusions

En-bloc resection and endoprosthetic reconstruction of secondary malignancies of the proximal femur with the modular devices used showed reliable results with respect to implant survival, local tumor and pain control even in advanced tumor stages as represented in this cohort. The use of a Trevira tube did not avoid dislocation or increase functional outcome in patients with severe soft tissue damage.

## Figures and Tables

**Figure 1 jcm-09-00758-f001:**
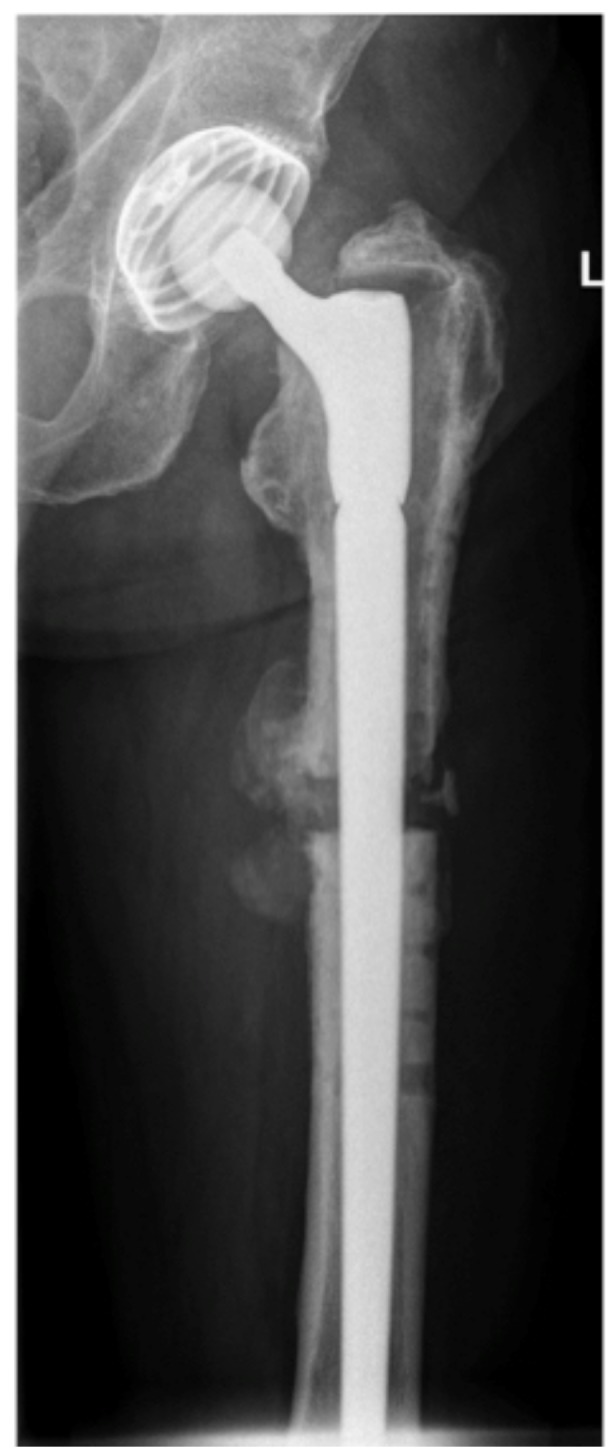
Anteroposterior (a.-p.) view of a 70-years old man that presented with a loosened THA. Thirty-five months before index operation, conversion of a failed osteosynthesis by a plate into a THA due to a pathologic fracture of a solitary metastasis of a renal cell carcinoma was performed (intralesional procedure with postoperative irradiation). Indication for revision of the THA was subsidence due to the non-integration of the cementless stem and instability related pain.

**Figure 2 jcm-09-00758-f002:**
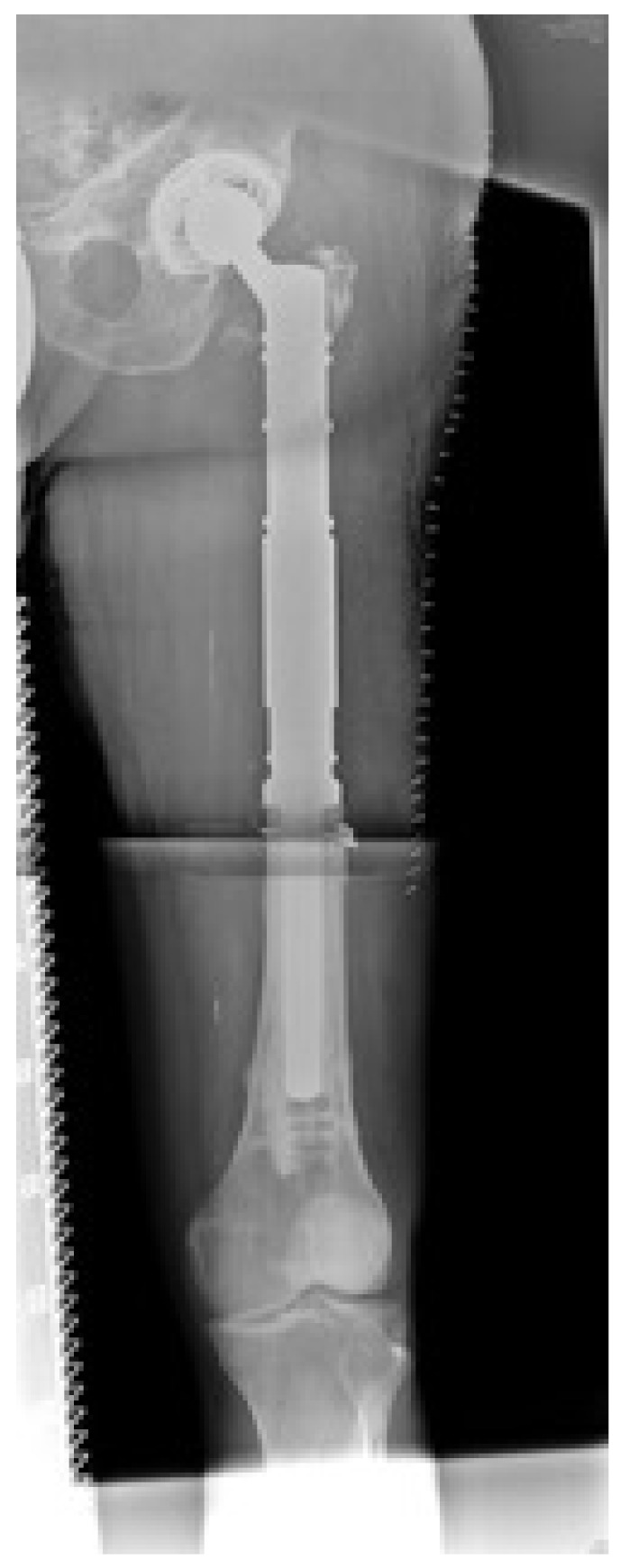
This X-ray (.-p. view) shows the status 13 months after resection of the proximal femur (former metastasis) and curettage of the distal part (intralesional procedure, histopathologically no tumor vitality after saturated initial irradiation). The Karnofsky index was 80% and the Musculoskeletal Tumor Society score (MSTS score) was 90% at the last follow up. The patient is still alive.

**Figure 3 jcm-09-00758-f003:**
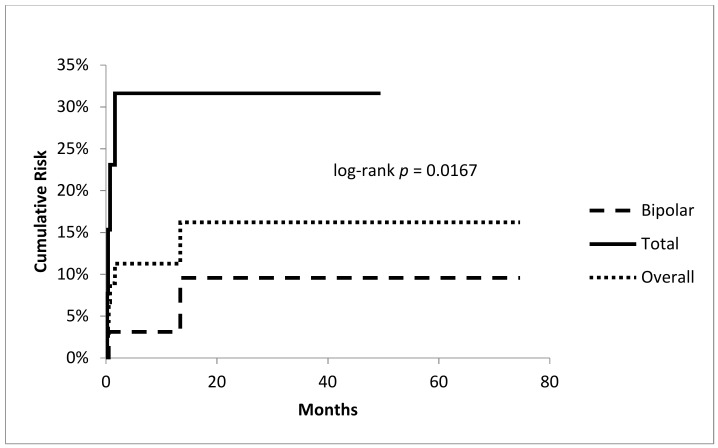
Cumulative risk of dislocation: overall rate and comparison between bipolar vs. total arthroplasties.

**Figure 4 jcm-09-00758-f004:**
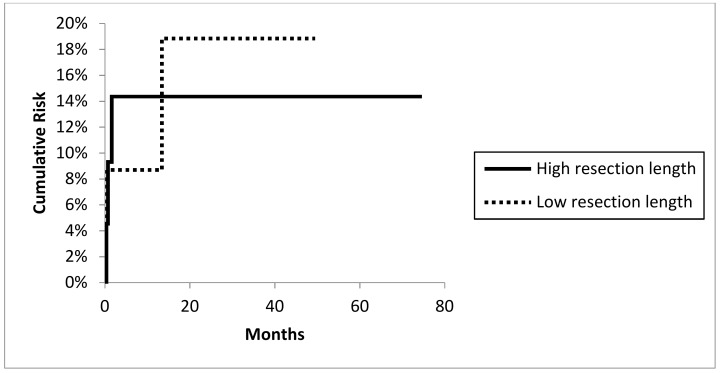
Cumulative risk of dislocation: higher or lower level than medium resection length 14.54 (± 4.25 SD; range 7–25) cm and dislocation risk (log-rank *p* = 0.9522).

**Figure 5 jcm-09-00758-f005:**
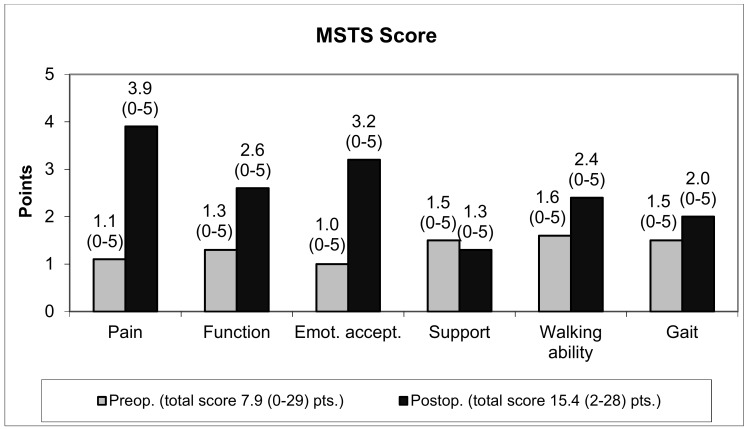
Functional outcome—MSTS score.

**Figure 6 jcm-09-00758-f006:**
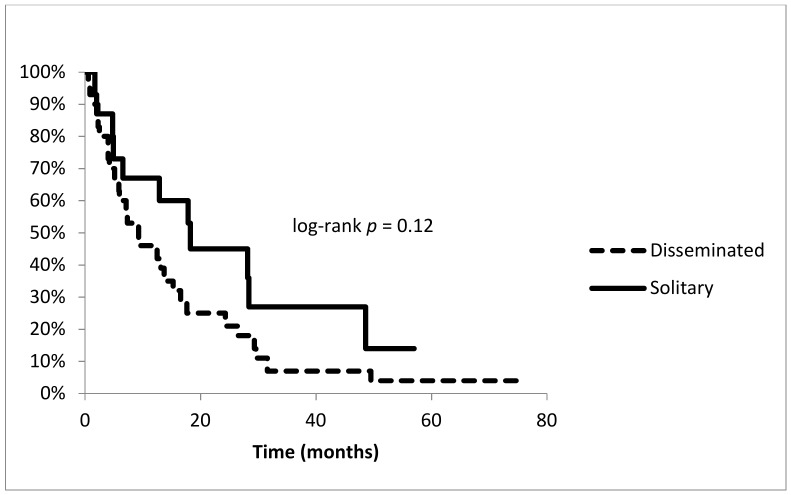
Survival of patients: comparison between solitary vs. disseminated disease at the time of operation.

**Figure 7 jcm-09-00758-f007:**
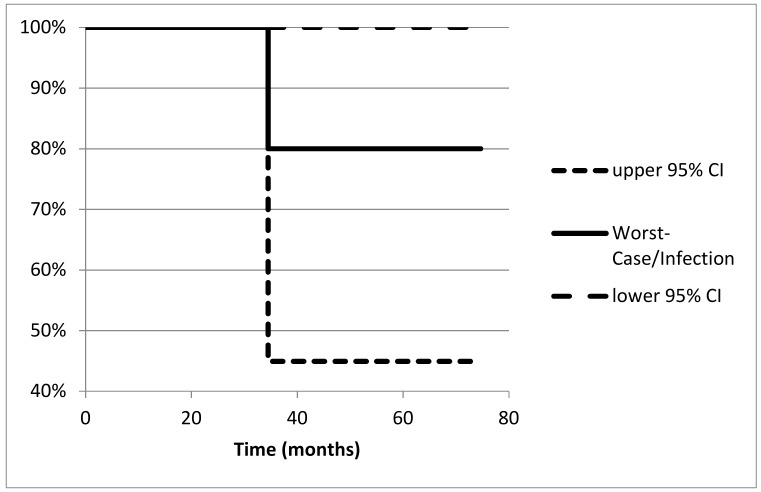
Survival of the implant: worst-case situation equates infection as failure reason (*n* = 1).

**Table 1 jcm-09-00758-t001:** Diagnosis and staging at the time of index operation.

Diagnosis and Staging at Index Operation	Number of Patients (45), Solitary/Disseminated
Bronchial Carcinoma	9 (5/4)
Leiomyosarcoma	1 (1/0)
Liposarcoma	1 (0/1)
Melanoma	1 (0/1)
Breast Carcinoma	13 (1/12)
Plasmacytoma	1 (0/1)
Oral Carcinoma	2 (0/2)
Renal Carcinoma	9 (4/5)
Paraganglioma	2 (0/2)
Pheochromocytoma	1 (1/0)
Prostate Cancer	1 (1/0)
Colorectal Cancer	1 (1/0)
Thyroidal Carcinoma	1 (0/1)
Urothelial Carcinoma	2 (1/1)

**Table 2 jcm-09-00758-t002:** Indication for operative treatment.

Indication for Operative Treatment	No. of Cases (*n* = 45)
Impending Fracture ^1^	23
Impending Fracture/Loosening after THA ^2^	1
Pathologic Fracture	17
Pathologic Fracture after Osteosynthesis ^3^	4

^1^ Mean Mirels Score (*n* = 23): 9 (7–11) pts. ^2^ 36 months after primary THA. ^3^ Proximal femoral nail (*n* = 3), double plate (*n* = 1); failure of osteosynthesis after a mean of 3.7 (0.1–8.6) months.

**Table 3 jcm-09-00758-t003:** Surgical margins and histopathological outcome.

	Surgical Margins/Histopathology		Number
Planned Surgical Procedure	Extralesional (R0)	Intralesional (R1/2)	
En-bloc Resection	37	4/1	42
Curettage Distally	0	2/1	3
Total	37	8	45

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
