# Peer review of "En-Bloc Resection of Metastases of the Proximal Femur and Reconstruction by Modular Arthroplasty is Not Only Justified in Patients with a Curative Treatment Option—An Observational Study of a Consecutive Series of 45 Patients"

_jcm, 2020, doi:10.3390/jcm9030758_

Round 1

Reviewer 1 Report

The authors describe the survivorship data of the implant, clinical outcomes and complication rates of en-bloc resection of metastasis performed in patients with advanced neoplastic disease in a retrospective study.

the structure of the study is original and the results are interesting in clinical practice.

The authors reported that survival in patients undergoing en-block resection with disseminated disease is similar to that of subjects with solitary metastasis. In patients with disseminated and advanced disease the intervention could have a palliative purpose. A control group of patients with bone metastasis at femur  but did not underwent to surgery should be retrospactively found.
The authors could  usefully compare the survival and the other clinical outcomes of these subjects with those of the population study.

The authors should discuss the study limitations and weaknesses.

Author Response

Thank you very much of your detailed remarks helping to improve the manuscript. We have made the following changes which are hopefully done in your sense.

1. A control group with conservative treatment is missing and should be compared with this cohort

Unfortunately, we haven't found current literature dealing with conservatively treated patients suffering of secondary bone neoplasm. As a great amount of the patients of the study presented with a fractured proximal femur, operative treatment was mandatory to get those patients mobile again. As it is known from fractures in elderly patients, especially fractures of the proximal femur near the hip, an early operative treatment and remobilization is directly correlated with a decrease of the one-year mortality.

The patients presented with impending fractures were classified according the Mirels score. Due to this classification and known fracture rates within 6 months despite other palliative treatment options other than surgery especially local radiotherapy, operative therapy is also mandatory due to the above mentioned reasons.

2. The limits and weaknesses of the paper should be discussed

A passage is added in the discussion chapter dealing with the limitations of this study.

Lines 260-266: 

‚Our study has also limitations. The comparatively small number of patients (n=45) is one disadvantage. Additionally, the follow-up period is limited due to the disease of the patients. Furthermore, a control group with patients treated by simple osteosynthesis, intralesional curettage and cemented primary stems or revision devices is missing. Nevertheless, there is adequate data available to compare the results of the current cohort and to draw the described conclusions. There are also limitations due to the study design as it is a retrospective study. Prospective randomized studies may be necessary in the future.‘

Kind regards

Reviewer 2 Report

This manuscript studies en-bloc resection of metastases of the proximal femur and the reconstruction by modular arthroplasty. This topic is of high interest to the community as there is limited consensus in the field regarding the surgical treatment of metastasis of the proximal femur. Overall, this study is very valuable as it add certain clarity to many questions in the field. Although only 45 patients were included, therefore limiting the conclusions which can be drawn, the study is well designed and results and conclusions are well presented.

Minor comments:

To enable readers to understand the story even better, figure legends should be improved. Please write a detailed legend for every figure which helps the reader to understand the figure without further reading of the text. Basically, state what is depicted and how the data have been generated/measured.

Fig 5 describes the patients' survival. As all patients are summarized here it would be interesting to separate early stage from late stage patients in two additional figure (potentially as supplement) to further describe how these subsets of patients are doing under the described treatments.

Also, negative data mentioned in the manuscript (line 153: resection/reconstruction length or radiotherapy showed no influence; line 166: Trevia tube did not influence..) should be added as supplementary figures to improve clarity on which parameters are positively affected and which onces are not.

Author Response

Dear Reviewer,

thank you very much of your detailed remarks helping to improve the manuscript. We have made the following changes which are hopefully done in your sense.

    1. Legends should be added to the figures in the manuscript

    The following legends are added:

    Lines103-107:

    Figure 1.

    A.-p. view of a 70-years old man presented with a loosened THA. 35 months before index operation, conversion of a failed osteosynthesis by a plate into a THA due to a pathologic fracture of a solitary metastasis of a renal cell carcinoma was performed (intralesional procedure with postoperative irradiation). Indication for revision THA was subsidence due to non-integration of the cementless stem and instability related pain.

    Lines110-112:

    Figure 2.

    This x-ray (.-p. view) shows the status 13 months after resection of the proximal femur (former metastasis) and curettage of the distal part (intralesional procedure, histopathologically no tumor vitality after saturated initial irradiation). The Karnofsky index was 80% and MSTS score 90% at latest follow up. The patient is still alive.

    Lines 170:

    Figure 3.

    Cumulative risk of dislocation: overall rate and comparison between bipolar vs. total arthroplasties

    Lines 173-75:

    Figure 4.

    Cumulative risk of dislocation: higher or lower level than medium resection length 14.54 (±4.25 SD; range 7-25) cm and dislocation risk (log-rank p=0.9522)

    Lines 189:

    Figure 5.

    Functional outcome-MSTS score

    Lines 209:

    Figure 6.

    Survival of patients: comparison between solitary vs. disseminated disease at time of operation

    Line 217

    Figure 7.

    Survival of implant: worst-case situation equates infection as failure reason (n=1)

    1. Additional supplemented figures should be added to Figure 5 (new numbering Figure no.6)

    The figure with all three lines included (disseminated, solitary and overall survival) may be somehow confusing. A figure without the overall survival is more impressive and may illustrate a tendency but not significantly improved outcome towards the solitary group. But as the main topic of the paper is the general benefit of a more radical way, even the disseminated patients may profit from this and has been shown due to the non-significant situation. Therefore, it may be better to give a figure without the overall situation which is further described in detail in the text. The surgical algorithm is equal in both groups.

    1. Additional supplemented figures should be added to the given data in the results chapter

    The numbering of the figures has been adapted according to the order of appearance in the text, we have added one additional figure.

    Line 162-63 ‘Resection length/reconstruction length showed no influence on dislocation rate.’

    A supplementary figure is attached (lines 172-75).

    Lines 183-85: ‘The use of a Trevira tube did not influence functional outcome.’

    We have analyzed the MSTS score and found no statistical benefit in the patients with a Trevira tube for soft tissue attachment compared to the patients with simple bonding of the remaining soft tissues. Additional figures should include the MSTS score and its subitems. An amount of 4 further figures would be necessary. As 7 figures have already been included in the paper and no further information is given with additional figures, we would propose not to add those figures.

Kind regards